# Design and Testing of Effective Primers for Amplification of the *orf7* Gene of Phage WO Associated with *Andricus hakonensis*

**DOI:** 10.3390/insects12080713

**Published:** 2021-08-09

**Authors:** Cheng-Yuan Su, Dao-Hong Zhu, Xiao-Hui Yang

**Affiliations:** 1Laboratory of Insect Behavior and Evolutionary Ecology, College of Life Science and Technology, Central South University of Forestry & Technology (CSUFT), Changsha 410004, China; suchengyuanzy@163.com; 2College of Life Science, Hunan Normal University, Changsha 410081, China; xhyang@hunnu.edu.cn

**Keywords:** Cynipidae, *Andricus hakonensis*, multiple infections, phage WO, *Wolbachia*

## Abstract

**Simple Summary:**

*Andricus hakonensis* is thought to contain the most complex and diverse phage types known and should be an ideal model material for studying interactions among bacteriophages, bacteria and eukaryotes. As shown in previous studies, existing primers are not effective enough to amplify all virus groups in *A. hakonensis*. Based on a comprehensive analysis of all virus groups reported to date, we designed a relatively conservative primer for virus detection. This primer can accurately and efficiently detect the presence of phage WO in arthropod hosts. Using gene alignment, clear evidence was provided for the existence of hitherto unreported base deletions, which are an important cause of diversity in phage WO associated with *A. hakonensis*.

**Abstract:**

Phage WO was first characterized in *Wolbachia*, an obligate intracellular Rickettsiales known for its ability to regulate the reproduction of arthropod hosts. In this paper, we focus on the study of virus diversity in *Andricus hakonensis* and the development of highly effective primers. Based on the existing *Wolbachia* genome sequence, we designed primers (WO-TF and WO-TR) to amplify the full-length *orf7* gene of phage WO. Surprisingly, sequencing results showed a high abundance of other phage WO groups in *A. hakonensis*, in addition to the four groups previously identified. The results also showed that *A. hakonensis* contained most of the known types of *orf7* genes (I, III, IV, V and VI) and the level of diversity of harbored phage WO was very high. Therefore, we speculated that existing primers were not specific enough and that new primers for the detection of phage WO were needed. Based on the existing *orf7* gene sequence, we designed specific detection primers (WO-SUF and WO-SUR). Sequencing results showed that the primers effectively amplified all known types of phage WO. In addition to amplifying most of the known sequences, we also detected some new genotypes in *A. hakonensis* using the new primers. Importantly, all phage WO groups could be efficiently detected. Combined with the results of previous studies, our results suggest that *A. hakonensis* contains the largest number of phage types (up to 36 types). This study is novel in that it provides practical molecular evidence supporting base deletions, in addition to gene mutations and genetic recombination, as an important cause of phage WO diversity.

## 1. Introduction

*Wolbachia* are obligated intracellular symbionts that are found in many filarial nematodes and arthropods [1] and are considered to infect about half of arthropod species in the biosphere [2,3]. Following phylogenetic analysis based on the *Wolbachia* surface protein (*wsp*) gene and the conserved coding gene 16S ribosomal RNA, strains are classified into 18 different supergroups (A–R) [4]. The high prevalence of *Wolbachia* is closely related to its ability to induce a variety of phenotypes, from regulating the reproduction of its arthropod hosts through mechanisms such as cytoplasmic incompatibility [5], parthenogenesis [6], male-killing [7] and feminization [8], to mutualism in nematodes [1]. Bacteriophages are thought to be the most abundant group of organisms in the biosphere [9]. Phage WO, the bacteriophage of obligated intracellular symbionts *Wolbachia*, plays a significant role in *Wolbachia* genomic evolution. Several recent studies have confirmed that the phage WO gene segment is crucial for cytoplasmic incompatibility of *Drosophila melanogaster*, induced by the *Wolbachia* associated with it [10,11,12]. The putative capsid protein encoded *orf7* gene is often used to identify bacteriophage infections and for phylogenetic analysis. Polymerase chain reaction (PCR) amplification using the universal primers (WO-F and WO-R) has shown that the phage WO gene segment occurs in most cases of the parasitic A and B *Wolbachia* supergroups [13,14]. Moreover, 85% of phage-infected *Wolbachia* strains harbors less than two phage types [14,15]. To the best of our knowledge, and *ricus hakonensis* contains the largest number of phage types; phage WO detection using WOF/R primers revealed 27 phage types [16].

Gall wasps (Hymenoptera: Cynipidae) are a phytophagous population of the superfamily Cynipoidea that represent the most spectacular branch of gall-inducing insects, with about 1400 species described [17]. It contains twelve tribes, i.e., Cynipini, Aylacini, Eschatocerin, Diplolepidini, Pediaspidili, Qwaqwaiini, Paraulacini, Aulacideini, Phanacidini, Diastrophini, Ceroptresini and Synergini [17]. *A. hakonensis* of the tribe Cynipini is widely distributed in southern China and it induces galls on *Quercus fabri* buds and leaves, respectively. We collected them from five places in Anhui, Hunan, Zhejiang and Fujian provinces in early May [16]. Several studies have detected high rates of *Wolbachia* infections in diverse cynipid species [18,19,20,21,22,23,24]. Moreover, some cynipid species are infected with multiple *Wolbachia* strains [22].

The tripartite temperate bacteriophage WO, the intracellular bacterium *Wolbachia* and their eukaryotic hosts are an ideal model system for studying interactions among bacteriophages, bacteria and eukaryotes [25]. In the *Wolbachia* genome, phage WO is a uniquely described transposable element [26] and prophage genomic regions can occupy more than twenty percent of mobile DNA genes. Moreover, prophage regions play a dominant role in the absent/divergent genes between closely related *Wolbachia* strains [27]. The relationships among the tripartite arthropod–Wolbachia–phage WO system are extremely intricate. Researchers have tried to explore the relationship between phage WO and *Wolbachia* associated with insect species such as *Drosophila melanogaster*, *Drosophila simulans*, *Culex pipiens***,**
*Nasonia vitripennis*, *Ephestia kuehniella* and *Gryllus pennsylvanicus* [10,11,12]. No evidence of phylogenetic congruence between phage WO and *Wolbachia* suggesting that the cross infection of phage WO in different *Wolbachia strains* is relatively common [13,27,28]. The CG content and codon preferences of phage WO and its *Wolbachia* host are similar; all the evidence suggests that phage WO have coevolved within *Wolbachia* for more than 100 million years [28]. Therefore, there must be a complicated relationship among the tripartite arthropod–*Wolbachia*–phage WO, which is worth exploring with further research.

The distribution of phage WO and the number of phage types in host arthropods may be greater than current estimates, as the primers WO-F and WO-R used to detect the presence of phage WO are not degenerate enough to detect every *orf7* gene variant. For instance, *D. simulans* was initially reported to have a single phage WO haplotype [14], but the genome sequence of the *Wolbachia* strain *w*Ri contains four confirmed prophage copies, three of which are unique [29]. Therefore, it is necessary to develop new detection primers of phage WO. By synthetically analyzing the published genome sequences of *Wolbachia*, we designed specific primers to amplify the full-length sequence of the phage *orf7* gene. With our newly designed primers (WO-TF and WO-TR), we detected viral *orf7* gene sequences in *A. hakonensis* that were not detectable using general WO-F and WO-R primers. The results showed that the gall wasp *A. hakonensis* contained most of the known *orf7* gene types (I, III, IV, V, VI and VII) and displayed a very high level of phage WO diversity, which is inconsistent with previous results [14,15]. Given the results we obtained, we decided to develop more versatile primers for the detection of phage WO. The experimental results show that our primers WO-SUF and WO-SUR are more effective than the conventional primers WO-F and WO-R in detecting the bacteriophage WO.

## 2. Materials and Methods

### 2.1. Insects and Total Genomic DNA Extraction

Galls of *A. hakonensis* were collected in early May between 2012 and 2021 from four provinces in southern China. All galls were reared under the appropriate humidity conditions in the zoology laboratory of Central South University of Forestry and Technology (CSUFT) and adult emergence was observed daily. Within 4 days of emergence, adults were directly preserved in 99.5% ethanol at −80 °C until DNA extraction. Adult gall wasps used for RNA extraction were preserved directly in liquid nitrogen.

Adult *A. hakonensis* were randomly selected for each location and all the samples were washed twice with sterile water before DNA extraction. Total genomic DNA was extracted from each insect using the phenol–chloroform extraction and ethanol precipitation method as previously described [21]. Suitably dried DNA samples were immediately resuspended in 50 µL of sterile ddH_2_O and deposited in 4 °C refrigerator for later use. We evaluated the quality of each *A. hakonensi* template by PCR using the nuclear ribosomal DNA internal transcribed spacer 2 (rDNA ITS2) gene [30]. Low quality templates were disposed responsibly to ensure the accuracy of infection rates.

### 2.2. PCR and Sequencing

All the samples were first screened for *Wolbachia* infection using the universal primers 81F and 691R (to amplify about 600 bp of *Wolbachia* surface protein gene) (Table 1) [31]. Phage WO of *Wolbachia* was then detected using the universal primers WO-F and WO-R (Table 1) to amplify about 350 bp of the Phage WO capsid protein gene *orf7* [28]. A blank control (ddH_2_O) was used for all amplifications. The reaction mixture was composed of 0.5 µL of each primer, 1 µL of template DNA, 0.2 µL of Pyrobest DNA Polymerase (5 U/μL)(Takara Biomedical Technology (Beijing) Co., Ltd, Dalian, China), 2.5 µL of 10×Pyrobest Buffer II, 2 µL of dNTP Mixture (2.5 mM each) and 18.3 µL of sterile ddH_2_O. The cycling conditions were as follows: 98 °C for 120 s, 30–40 cycles of 98 °C for 10 s, 50–57 °C for 30 s and 68 °C for 1–2 min, using a C1000 Touch Thermal Cycler (Bio-Rad, Hercules, CA, USA).

The PCR products (2.5 μL) were run on a 0.8% Invitrogen agarose gel electrophoresis with 0.3 μL of 10 × Loading Buffer, under 1× TBE buffer. The gels were soaked in GelRed diluent for 30 min and a picture was taken using a gel imaging system (Bio-Rad, Hercules, CA, USA). All positive PCR products were then purified using an Agarose Gel DNA Extraction Kit (Takara Biomedical Technology (Beijing) Co., Ltd.). The purified PCR products were sequenced using the corresponding PCR primers. If the initial sequencing results showed multiple peaks, this sample was considered as multiple phage WO infections. The multiple phage WO infections PCR products were then purified and ligated into the Mighty TA-cloning Reagent Set for PrimeSTAR pMD20-T vector (Takara Biomedical Technology (Beijing) Co., Ltd.) following the manufacturer’s procedures. Firstly, for each adult, 10–15 positive colonies were randomly selected and cultured in a lysogeny broth medium supplemented with ampicillin. Secondly, plasmids were carefully extracted. Finally, we sequenced the plasmid in both directions with two primers (M13F and M13R) using an ABI 3730XL DNA Sequencer (Applied Biosystems, Foster City, CA, USA) at Icongene Co., Ltd. (Wuhan, China).

### 2.3. Raw Sequence Treatment

Firstly, we performed the sequence homology analysis using the online BLAST program. Secondly, genetic distances between all established sequence pairs were calculated using MEGA 7.0 under Kimura 2-parameter distance models [32]. All the sequencing results in the current study were deposited in GenBank under the following accession numbers: MZ687122-MZ687127; MZ687129-MZ687140; MZ687810-MZ687824.

### 2.4. Phylogenetic Analyses

The maximum likelihood (ML) phylogenetic tree was carried out using IQ-Tree 2.1.1 via the online CIPRES Science Gateway V. 3.3 using the general time-reversible (GTR) model, in which the invariant sites and gamma distribution were estimated (GTR + I + G) [33]. Model Test v3.7 was used to choose the best evolutionary model. Bootstrap values were generated from 10,000 bootstrap replicates.

### 2.5. RNA Extraction

We assessed three samples composed of pooled females and three samples of pooled males. The number of individuals in each sample ranged from 10 to 25 according to body size (approximately 10–12 females per sample and 18–25 males per sample). Before RNA extraction, adult *A. hakonensis* were washed twice with sterile water. We used RNAiso Plus (Takara Biomedical Technology Co., Ltd.) to extract total RNA for each sample and suitably dried RNA samples were resuspended in 50 µL of RNase-free ddH_2_O. The concentration and purity of each RNA sample were assessed by measuring their optical density (OD) values using a Thermo Scientific NanoDrop One. PCR tests were performed on all RNA samples using the mitochondrial cytochrome c oxidase 1 (*cox1*) gene [34] and nuclear ribosomal DNA internal transcribed spacer 2 gene [30] as negative controls to affirm that the RNA samples did not contain high levels of *A. hakonensis* genomic DNA contamination. cDNA was then synthesized using a PrimeScript RT Reagent Kit and remaining genomic DNA contamination was removed using gDNA Eraser (Takara Biomedical Technology Co., Ltd.). The reaction mixture was incubated at 37–50 °C for 40–90 min using a C1000 Touch Thermal Cycler.

## 3. Results

### 3.1. Acquisition and Typing of the Full-Length Sequence of the Phage WO orf7 Gene

In order to obtain the full-length sequence of the phage WO *orf7* gene, we first used inverse PCR. Unfortunately, we were unable to obtain the target gene sequence despite trying various combinations of restriction enzymes. Owing to the failure of the inverse PCR experiment, we were only able to obtain specific viral amplification primers by comparative analysis of virus-related gene sequences from published *Wolbachia* genomes (Appendix A). We downloaded *Wolbachia* genomes containing the viral gene fragments and obtained the specific primers WO-TF and WO-TR (Table 1) by comparing the genes in the conserved region. We obtained a total of 32 positive clones that could be divided into four types, from WOAha-1T to WOAha-4T. Surprisingly, phylogeny results showed that *A. hakonensis* was infected with a large number of other phage WO groups, in addition to the four groups identified using WO-F and WO-R (Figure 1). In fact, only WOAha-4T genotypes had previously been detected, whereas WOAha-1T, WOAha-2T and WOAha-3T are newly discovered genotypes [16]. Phylogeny results based on WO-TF and WO-TR showed that three genotypes were undetectable using the primers WO-F/R. It is possible that these results may be due to amplification bias of the primers we used. However, it is indisputable that universal primers, such as WO-F and WO-R, cannot efficiently amplify all phage WO genotypes. Combined with results of the previously identified phage WO genotypes, the phylogeny results show that *A. hakonensis* contained most of the known *orf7* gene types (I, III, IV, V, VI and VII) (Figure 2). Most of the orf7 genotypic differences associated with *A. hakonensis* result in changes in protein sequences (Appendix A).

### 3.2. Design of Highly Effective Primers for virus Amplification

In view of our results, we speculate that existing primers are not specific enough to amplify all phage WO types. We found that primers WO-F and WO-R resulted in a large amount of nonspecific amplification, including around 350 bp, that could interfere with the measurement of infection rates. These results support the urgent need to design new detection primers for phage WO. Based on the existing *orf7* gene sequence, we designed specific detection primers (WO-SUF and WO-SUR) (Table 1) to obtain the target band of around 250 bp. The relative positions of all primers used in this article are shown in Appendix A. We obtained a total of 202 positive clones that could be divided into 19 types. In addition to amplifying most of the known sequences, we also detected some new genotypes (Figure 3). Most importantly, all phage WO groups could be efficiently detected. Thus, to the best of our knowledge, *A. hakonensis* contains the largest number of phage types (up to 36 types) (Figure 1, Figure 2 and Figure 3). We also applied the primers to ten other gall wasps, fifteen crickets, three locusts, two aphids and sixteen butterflies and were able to obtain specific amplification bands for all (Appendix A).

### 3.3. Base Deletions in orf7

Diverse phage WO types in *A. hakonensis* provided a perfect sample for obtaining evidence of genetic evolution. This article is novel in providing actual molecular evidence supporting that base deletions are also an important cause of phage WO diversity associated with *A. hakonensis*, in addition to gene mutations and genetic recombination, which have been reported previously [16]. When the gene sequences were aligned in the way that amino acids are encoded, we saw that all the base deletions caused amino acid deletions (Figure 4). The WOAha-33 deletion “TAAAAG” resulted in WOAha-7 and WOAha-8 and two base deletions were found among WOAha-26, WOAha-11, WOAha-14, WOAha-15 and WOAha-17.

### 3.4. A High Proportion of Phage WO Types Are Actively Transcribed in A. hakonensis

We verified the transcriptional activity of phage WO in *A. hakonensis* using reverse transcription PCR (RT-PCR). We obtained a total of 37 positive clones, which could be summarized as nine types belonging to WOAha-1/4/6/12/13/15/18/21 and WOAha-26 (Figure 5). These results strongly suggest that most phage WO in *A. hakonensis* are transcribed normally and are able to perform normal biological functions.

## 4. Discussion

Although bacteriophages are typically thought to be rare or absent in obligate intracellular bacteria [35,36], phage WO is widely distributed in a variety of *Wolbachia*-infected insect groups [13,14,27,37,38,39,40]. Moreover, multiple phage infections (*Wolbachia* strains containing more than one phage type) can be found in several *Wolbachia* strains [14,40].

Our phylogeny results reveal a high level of phage WO diversity in *A. hakonensis* and show that phage WO can be divided into four groups [16]. Phylogeny results based on WO-TF and WO-TR show that most of the genotypes were undetectable using the original universal primer WO-F /R, indicating that existing primers are not specific enough to amplify all phage WO types. Therefore, there is an urgent need to design a new detection primer for phage WO. In *A. hakonensis*, the primers WO-SUF and WO-SUR efficiently detected all known phage WO groups, demonstrating the high efficiency of the primers. These results confirm that *A. hakonensis* contains the largest number of phage types (up to 36 types), which can be divided into six groups (I, III, IV, V, VI and VII). However, phage-infected *Wolbachia* strains are thought to harbor low numbers of phage types (with 85% harboring only one or two phage WO types) [14,15]. One explanation for the large number of phage types in *A. hakonensis* is genetic mutations. The results of Zhu et al. (2021) [16] strongly suggest that intragenic recombination was an important evolutionary dynamic promoting a high level of phage WO diversity in gall wasps. Moreover, the present study is novel in providing practical molecular evidence supporting the fact that base deletions are also an important cause of phage WO diversity associated with *A. hakonensis*. We also applied the primers to the other 46 species of insects, belonging to Hymenoptera, Orthoptera, Hemiptera and Lepidoptera, and all of them were able to obtain specific amplification bands. The results fully demonstrated the specificity and effectiveness of the newly designed primers. Phage WO phylogenetic relationships inferred from only a portion of the *orf7* gene sequence may not be highly reliable. This is demonstrated by our comparison of WOAha-1T (1089 bp) with WOAha-2T (1068 bp). The first 1016 bases of the sequences of these two genotypes are almost identical, but the remainder of the sequence is completely different (Appendix A). Given the frequency of intragenic recombination, more genetic information is needed to accurately reconstruct phylogenetic relationships of phage WO.

## 5. Conclusions

Given the results of our experiments, the distribution and types of phage WO in host arthropods may be greater than previous estimates, as the primers used to screen for the presence of phage WO are not sufficiently degenerate to detect all *orf7* variants. Our newly designed specific primers (WO-SUF and WO-SUR) may be more suitable for detecting the presence of phage WO in arthropods. Most phage types in *A. hakonensis* are transcribed normally, which reveals that they are able to perform normal biological functions. Therefore, *A. hakonensis* may be an ideal model system for studying interactions among eukaryotes, bacteria and bacteriophages.

## Figures and Tables

**Figure 1 insects-12-00713-f001:**
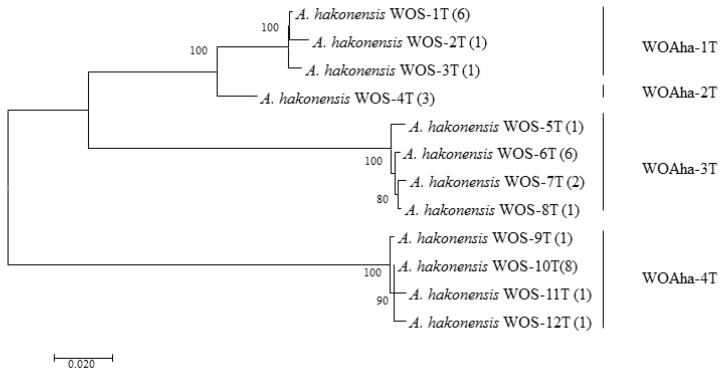
Maximum likelihood tree for phage WO types of *A. hakonensis* based on full length of *orf7* sequences. The numbers in parentheses indicate the number of sequences obtained. WOS-xT refers to the serial number. Phage WO types are shown on the right. Numbers above and below branches are bootstrap values computed from 1000 replications.

**Figure 2 insects-12-00713-f002:**
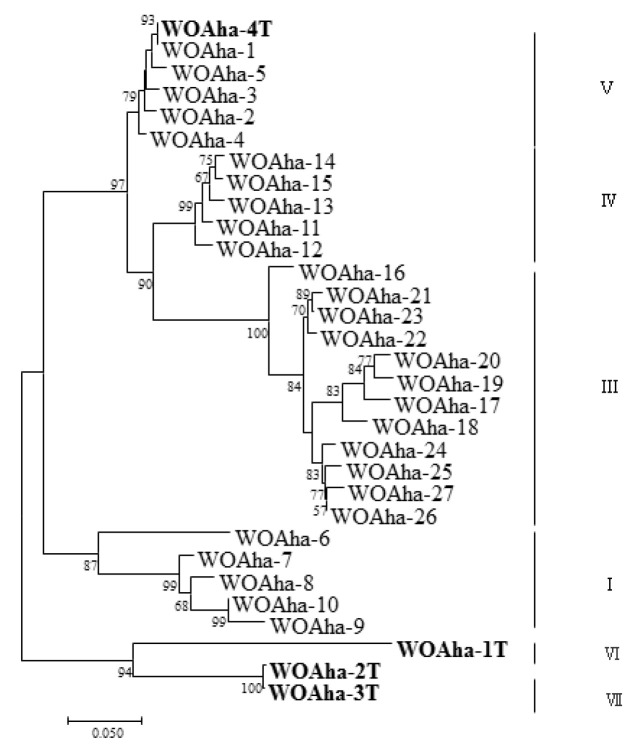
Maximum likelihood tree for phage WO types of *A. hakonensis* based on *orf7* sequences. WOAha-X refers to the phage WO types, “T” indicate the full length sequence of *orf7*. Numbers above and below branches are bootstrap values computed from 1000 replications.

**Figure 3 insects-12-00713-f003:**
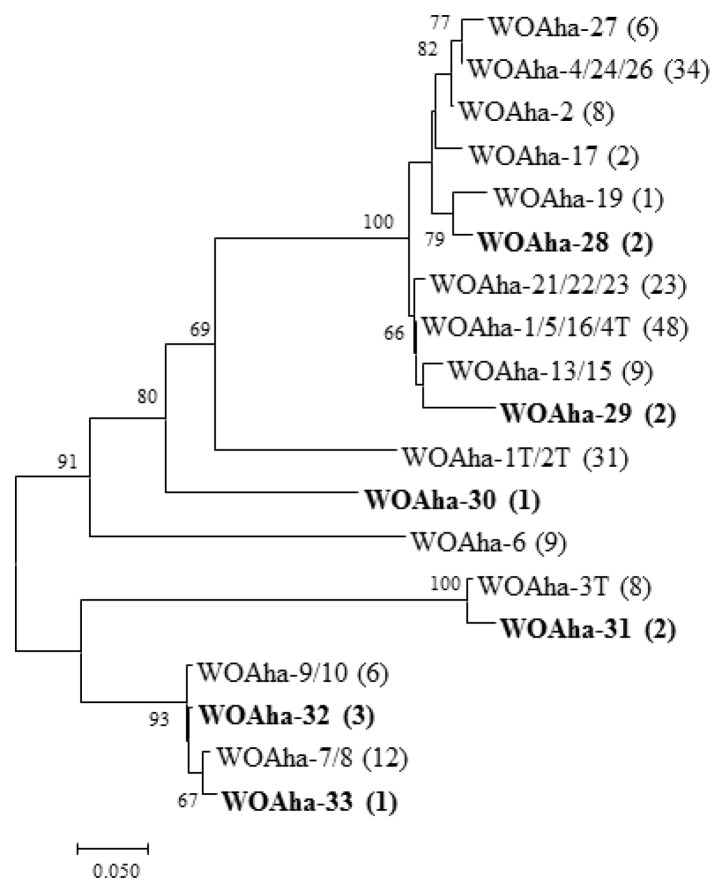
Maximum likelihood tree for phage WO types of *A. hakonensis* based on *orf7* sequences obtain with new detection primers. The numbers in parentheses indicate the number of sequences obtained.

**Figure 4 insects-12-00713-f004:**
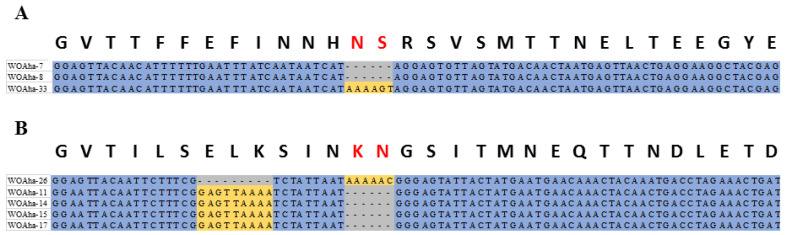
Bases deletions of the orf7 gene (**A**) WOAha-33 deletion “TAAAAG” resulting in WOAha-7 and WOAha-8, and (**B**) Bases deletions among WOAha-26, WOAha-11, WOAha-14, WOAha-15, and WOAha-17.

**Figure 5 insects-12-00713-f005:**
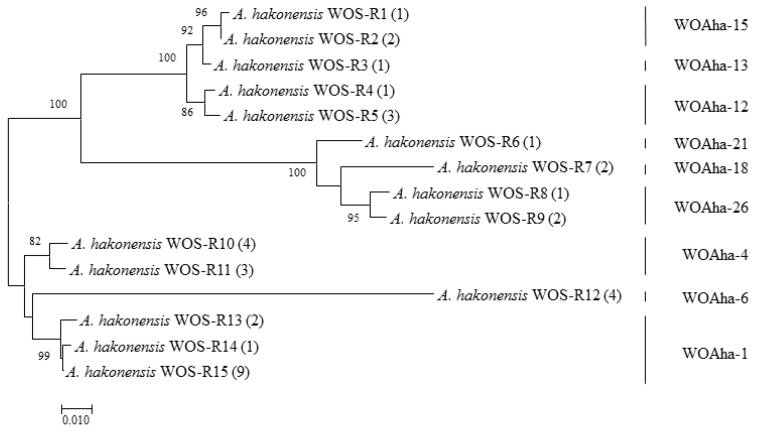
Maximum likelihood tree actively transcribed phage WO types of *A. hakonensis* based on *orf7* sequences using WO-F; WO-R primers. WOS-Rx refers to the serial number. The numbers in parentheses indicate the number of sequences obtained. Phage WO types are shown on the right. Numbers above and below branches are bootstrap values computed from 1000 replications.

**Table 1 insects-12-00713-t001:** Primers used in this study.

	Gene	Primer	Primer Sequence	Annealing Temperature	References
Phage WO	*orf7*	WO-F WO-R	5′-CCC ACA TGA GCC AAT GAC GTC TG-3′ 5′-CGT TCG CTC TGC AAG TAA CTC CAT TAA AAC-3′	57 °C	Masui et al., 2000
		WO-TF WO-TR	5′-CTTGGCTACYGTTATYTGRTCTCTTG-3′ 5′-TCYGASGTTACTRCHAATCAAGAGG-3′	59 °C	This study
		WO-SUF: WO-SUR:	5′-ARGCAGGKCTWTATTTTGG-3′ 5′-ATTCTTCTATYTTYYCTGGCA-3′	50 °C	This study
*Wolbachia*	*wsp*	81F 691R	5′-TGG TCC AAT AAG TGA TGA AGA AAC-3′ 5′-AAA AAT TAA ACG CTA CTC CA-3′	52 °C	Zhou et al.,1998
Insect	ITS2	ITS2F ITS2R	5′-TGT GAA CTG CAG GAC ACA TG-3′ 5′-AAT GCT TAA ATT TAG GGG GTA-3′	54 °C	Partensky and Garczarek, 2011
	*cox1*	LCO-1490 HCO-2198	5′-GGTCAACAAATCATAAAGATATTGG-3′ 5′-TAAACTTCAGGGTGACCAAAAATCA-3′	55 °C	Dyer et al., 2011

## Data Availability

All the sequencing results in the current study were deposited in GenBank under the following accession numbers: MZ687122-MZ687127; MZ687129-MZ687140; MZ687810-MZ687824.

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
