# Peer review of "Design and Testing of Effective Primers for Amplification of the orf7 Gene of Phage WO Associated with Andricus hakonensis"

_insects, 2021, doi:10.3390/insects12080713_

Round 1
Reviewer 1 Report
The authors describe a new pair of universal PCR primers to amplify a short fragment of phage WO in a Wolbachia-infected gall wasp.
General comments
- the reported results and conclusions heavily depend on “previous work” (ref 16) (lines 53-56; 89-83; 198-199; 261-263; 270-271), but the paper appears not available at present.
- the authors refer to results not shown (lines191-193) to underscore the suitability of the primers proposed in this paper.
- It is unclear which specific part orf7 is amplified by the newly design primers; at least a figure showing orf7 and the relative positions of all primers used is needed.
- phylogenetic analysis were performed on DNA sequences only. It is unclear how the variations in the coding sequence affect the protein sequence (phenotype). The authors should analyse this as well.
- Supplementary data (Table S1, Fig S1) are missing, but needed to understand the results.
Specific comments
Introduction
• the relevance of Andricus haconensis is not mentioned. Only in line 82, we get to know it is a gall wasp.
• line 88-92: the general paragraph on gall wasps at the end of the discussion should be moved earlier in the discussion
• any background information on WO phage orf7 (minor capsid gene) is missing.
• line 53-56 refers to own results (ref 16) of which apparently are not published yet.
• line 80-83 refers to the same results, but the reference (ref 16) is missing, creating the expectation that the results will be described in this manuscript.
Materials and method
• line 101: correct (RNA or DNA extraction)
• line 127: mention which vector
• line 135: add reference to MEGA 7.0
• lines 145-147: the sample size description is confusing. The two sentences contradict each other
• line 154: change “cytochrome coxidase1” into “cytochrome c oxidase 1”
Results
• line 161-163: “reverse PCR” is not commonly used as a method name; most probably refers to “reverse transcription PCR”. The first reference to “reverse transcription PCR (RT-PCR)” is in lines 204-205. The restriction analysis is not mentioned in Materials and methods.
• lines 173-174: the genotypes undetectable with primers WO-F/R are represented as “approximately 65,6%.” It would be more appropriate to mention the exact numbers since only 32 clones were obtained.
• lines 177-179: the authors point to “previously identified phage WO genotypes” but do not provide the reference (own unpublished paper 16?)
• lines 182-184: it is unclear what the authors mean with nonspecific PCR products that could interfere with “the measurement of infection rates”.
• lines 191-193: the authors refer to specific PCR products obtained from 46 other insect species as “results not shown”. The new primers would prove suitable if the results were shown!
• lines 199-200: “the gene sequences were aligned in the way that amino acids are encoded”: the reading frame has to be added to Fig 4.
Discussion
• Line 265: the references to SARS-CoV (refs 40, 41) which is an unrelated eukaryotic virus are inappropriate to underline the importance of deletions in phage WO sequence variation.
Figures & tables
Fig 2: some bootstrap values are above, some below the branches (the legend refers to numbers above the branches)
Table 1: primer WO-F and WO-R names (WOF; WOR) are inconsistent
Author Response
<<Responses to Reviewer 1 >>
Our response follows the comment.
--------
General comments
- the reported results and conclusions heavily depend on “previous work” (ref 16) (lines 53-56; 89-83; 198-199; 261-263; 270-271), but the paper appears not available at present.
- The article is available now : Zhu D-H, Su C-Y, Yang X-H and Abe Y (2021) A Case of Intragenic Recombination Dramatically Impacting the Phage WO Genetic Diversity in Gall Wasps. Microbiol. 12:694115. doi: 10.3389/fmicb.2021.694115
- the authors refer to results not shown (lines191-193) to underscore the suitability of the primers proposed in this paper.
- We have supplemented the corresponding PCR products of the orf7 gene fragments using WO-SUF and WO-SUR.
- It is unclear which specific part orf7 is amplified by the newly design primers; at least a figure showing orf7 and the relative positions of all primers used is needed.
- The relative positions of all primers used in this article as shown in the figure below.
- phylogenetic analysis were performed on DNA sequences only. It is unclear how the variations in the coding sequence affect the protein sequence (phenotype). The authors should analyse this as well.
- Most of the orf7 genotypic differences associated with A.hakonensis result in changes in protein sequences. The phylogenetic analysis based on the protein sequence as shown in the figure below.
- Supplementary data (Table S1, Fig S1) are missing, but needed to understand the results.
- We have provide related data accordingly.
Specific comments
Introduction
- the relevance of Andricus haconensis is not mentioned. Only in line 82, we get to know it is a gall wasp.
- We have rewrite as “ hakonensis of the tribe Cynipini is widely distributed in southern China, which induce galls on host leaves and buds, respectively. We collected them from five places in Anhui, Hunan, Zhejiang and Fujian provinces in early may.”
- line 88-92: the general paragraph on gall wasps at the end of the discussion should be moved earlier in the discussion
- We have do it accordingly.
- any background information on WO phage orf7 (minor capsid gene) is missing.
- We have rewrite as “The putative capsid protein encoded orf7 gene, is often used to identify bacteriophage infections and for phylogenetic analysis.”
- line 53-56 refers to own results (ref 16) of which apparently are not published yet.
- The article is available now : Zhu D-H, Su C-Y, Yang X-H and Abe Y (2021) A Case of Intragenic Recombination Dramatically Impacting the Phage WO Genetic Diversity in Gall Wasps. Microbiol. 12:694115. doi: 10.3389/fmicb.2021.694115
- line 80-83 refers to the same results, but the reference (ref 16) is missing, creating the expectation that the results will be described in this manuscript.
- The results were described in this manuscript (Fig. 1, 2).
Materials and method
- line 101: correct (RNA or DNA extraction)
- We have modified it as RNA extraction.
- line 127: mention which vector
- We have modified it as “Mighty TA-cloning Reagent Set for PrimeSTAR pMD20-T vector (Takara Biomedical Technology Co., Ltd.)”
- line 135: add reference to MEGA 7.0
- We have do it accordingly.
Sudhir Kumar, Glen Stecher, Koichiro Tamura, MEGA7: Molecular Evolutionary Genetics Analysis Version 7.0 for Bigger Datasets, Molecular Biology and Evolution, Volume 33, Issue 7, July 2016, Pages 1870–1874.
- lines 145-147: the sample size description is confusing. The two sentences contradict each other
- Because the gall wasp was so small, we extracted RNA by mixing samples. We have modified “gall wasps”as “samples”
- line 154: change “cytochrome coxidase1” into “cytochrome c oxidase 1”
- We have do it accordingly.
Results
- line 161-163: “reverse PCR” is not commonly used as a method name; most probably refers to “reverse transcription PCR”. The first reference to “reverse transcription PCR (RT-PCR)” is in lines 204-205. The restriction analysis is not mentioned in Materials and methods.
- We have modified “reverse PCR”as “inverse PCR”
The principle of inverse PCR is as follows:
- lines 173-174: the genotypes undetectable with primers WO-F/R are represented as “approximately 65,6%.” It would be more appropriate to mention the exact numbers since only 32 clones were obtained.
- We have corrected it as “three”
- lines 177-179: the authors point to “previously identified phage WO genotypes” but do not provide the reference (own unpublished paper 16?)
- That's true.
- lines 182-184: it is unclear what the authors mean with nonspecific PCR products that could interfere with “the measurement of infection rates”.
- We used PCR to detect whether the species was infected. When we use the original WO-F;WO-R to detect the WO phage infection, bands of similar size to the target band are often obtained. Sequencing results showed that these were non-specific amplifications. But when we look at infection rates, we don't always sequence all the samples, so the non-specific amplifications will affect the measurement of infection rates.
- lines 191-193: the authors refer to specific PCR products obtained from 46 other insect species as “results not shown”. The new primers would prove suitable if the results were shown!
- We have supplemented the corresponding PCR products of the orf7 gene fragments using WO-SUF and WO-SUR.
- lines 199-200: “the gene sequences were aligned in the way that amino acids are encoded”: the reading frame has to be added to Fig 4.
- We have do it accordingly.
Discussion
- Line 265: the references to SARS-CoV (refs 40, 41) which is an unrelated eukaryotic virus are inappropriate to underline the importance of deletions in phage WO sequence variation.
- We deleted it.
Figures & tables
Fig 2: some bootstrap values are above, some below the branches (the legend refers to numbers above the branches)
- We have corrected it as “Numbers above and below branches are bootstrap values computed from1 000 replications.”
Table 1: primer WO-F and WO-R names (WOF; WOR) are inconsistent
- We have corrected it.

Reviewer 2 Report
The authors report a novel study providing practical molecular evidence supporting base deletions, in addition to gene mutations and genetic recombination, as an important cause of phage WO diversity. It is a type of phage that infects the strictly intracellular bacterium Wolbachia, ubiquitous in arthropods.
Although the study is of clear interest to the phage-researcher audience, the Figures should be highly improved due to their quite low resolution. In addition, more Figures, such as plots, would add value to the work already developed by the authors.
The Discussion section of the manuscript must also be improved, since as it is now it looks more like a Conclusions section and not a Discussion.
Author Response
<<Responses to Reviewer 2>>
We are very happy to read the conclusion of your review.
Our response follows the comment.
Comments and Suggestions for Authors
The authors report a novel study providing practical molecular evidence supporting base deletions, in addition to gene mutations and genetic recombination, as an important cause of phage WO diversity. It is a type of phage that infects the strictly intracellular bacterium Wolbachia, ubiquitous in arthropods.
Although the study is of clear interest to the phage-researcher audience, the Figures should be highly improved due to their quite low resolution. In addition, more Figures, such as plots, would add value to the work already developed by the authors.
- We have do it accordingly.
Figure 4 is corrected (aligned using the amino acid sequence); supplemented fig s2-4.
The Discussion section of the manuscript must also be improved, since as it is now it looks more like a Conclusions section and not a Discussion.
- We have do it accordingly.

Round 2
Reviewer 1 Report
A major shortcoming of the first version was the referencing to own unpublished work. This has been addressed in the 2nd version by the publication of the work in question. Also the missing supplementary data have been made available.
Some minor remaining comments:
• line 159: is still ambiguous: “We assessed six samples from three female and three male samples”: I suppose you mean “We assessed three samples composed of pooled females and three samples of pooled males”?
• The sentence “The relative positions of all primers used in this article as shown in the figure S4.” has been added twice, in lines 98-99 (Introduction) and in lines 203-204 (Results section). The latter is more appropriate and sufficient.
• Some items that were addressed according to the authors' reply letter were unchanged in the revised manuscript:
- line 154: “cytochrome coxidase” must be split into “cytochrome c oxidase”
- line 161-163 / line 175-177 in version 2: “reverse PCR” is not commonly used as a method name; most probably refers to “reverse transcription PCR”. The first reference to “reverse transcription PCR (RT-PCR)” is in lines 204-205. The restriction analysis is not mentioned in Materials and methods.
The authors replied it would have been changed into "inverse PCR", which is even more obscure. Can the experimental approach that failed to yield the ORF7 full length sequence be cleared out?
• The sentence “The relative positions of all primers used in this article as shown in the figure S4.” has been added twice, in lines 98-99 (Introduction) and in lines 203-204 (Results section). The latter is more appropriate and sufficient.
Author Response
<<Responses to Reviewer 1 >>
--------
Comments and Suggestions for Authors
A major shortcoming of the first version was the referencing to own unpublished work. This has been addressed in the 2nd version by the publication of the work in question. Also the missing supplementary data have been made available.
Some minor remaining comments:
- line 159: is still ambiguous: “We assessed six samples from three female and three male samples”: I suppose you mean “We assessed three samples composed of pooled females and three samples of pooled males”?
- That's true. We have corrected it as “We assessed three samples composed of pooled females and three samples of pooled males”. Thank you very much.
- The sentence “The relative positions of all primers used in this article as shown in the figure S4.” has been added twice, in lines 98-99 (Introduction) and in lines 203-204 (Results section). The latter is more appropriate and sufficient.
- That's true. We deleted it.
- Some items that were addressed according to the authors' reply letter were unchanged in the revised manuscript:
line 154: “cytochrome coxidase” must be split into “cytochrome c oxidase”
- I'm very sorry. We have do it accordingly.
line 161-163 / line 175-177 in version 2: “reverse PCR” is not commonly used as a method name; most probably refers to “reverse transcription PCR”. The first reference to “reverse transcription PCR (RT-PCR)” is in lines 204-205. The restriction analysis is not mentioned in Materials and methods.
The authors replied it would have been changed into "inverse PCR", which is even more obscure. Can the experimental approach that failed to yield the ORF7 full length sequence be cleared out?
- I'm very sorry. We have do it accordingly.
The principle of inverse PCR is shown in the following figure:
Inverse PCR (also known as inverted or inside-out PCR) is used to amplify DNA sequences that flank one end of a known DNA sequence and for which no primers are available. Inverse PCR DNA involves digestion by a restriction enzyme of a preparation of DNA containing the known sequence and its flanking region. The individual restriction fragments (many thousands in the case of total mammalian genomic DNA) are converted into circles by intramolecular ligation, and the circularized DNA is then used as a template in PCR. The unknown sequence is amplified by two primers that bind specifically to the known sequence and point in opposite directions. The product of the amplification reaction is a linear DNA fragment containing a single site for the restriction enzyme originally used to digest the DNA. This site marks the junction between the previously cloned sequence and the flanking sequences. The size of the amplified fragment depends on the distribution of restriction sites within known and flanking DNA sequences.
We think that even though we didn't get the results, but this experimental method embodies our way of thinking. We want to represent the true process of how we solve the scientific problems. If you still feel it is not suitable, we can removing it.
- The sentence “The relative positions of all primers used in this article as shown in the figure S4.” has been added twice, in lines 98-99 (Introduction) and in lines 203-204 (Results section). The latter is more appropriate and sufficient.
- I'm very sorry. We have do it accordingly.
This manuscript is a resubmission of an earlier submission. The following is a list of the peer review reports and author responses from that submission.